# Use of Transabdominal Ultrasound for the detection of intra-peritoneal tumor engraftment and growth in mouse xenografts of epithelial ovarian cancer

Laura M. Chambers[1], Emily Esakov[2], Chad Braley[2], Mariam AlHilli[1], Chad Michener[1], Ofer Reizes[2]*

1 Division of Gynecologic Oncology, Obstetrics, Gynecology and Women's Health Institute, Cleveland Clinic, Cleveland, OH, United States of America, 2 Department of Cardiovascular and Molecular Medicine, Lerner Research Institute, Cleveland Clinic, Cleveland, OH, United States of America

* reizeso@ccf.org

**Data Availability Statement:** All relevant data are within the paper and its Supporting Information files.

## Abstract

### Objective

To evaluate intraperitoneal (IP) tumor engraftment, metastasis and growth in a pre-clinical murine epithelial ovarian cancer (EOC) model using both transabdominal ultrasound (TAUS) and bioluminescence *in vivo* imaging system (IVIS).

### Methods

Ten female C57Bl/6J mice at six weeks of age were included in this study. Five mice underwent IP injection of $5 \times 10^6$ ID8-luc cells (+ D- luciferin) and the remaining five mice underwent IP injection of ID8-VEGF cells. Monitoring of tumor growth and ascites was performed weekly starting at seven days post-injection until study endpoint. ID8-luc mice were monitored using both TAUS and IVIS, and ID8-VEGF mice underwent TAUS monitoring only. Individual tumor implant dimension and total tumor volume were calculated. Average luminescent intensity was calculated and reported per mouse abdomen. Tumor detection was confirmed by gross evaluation and histopathology. All data are presented as mean +/- standard deviation.

### Results

Overall, tumors were successfully detected in all ten mice using TAUS and IVIS, and tumor detection correlated with terminal endpoint histology/ H&E staining. For TAUS, the smallest confirmed tumor measurements were at seven days post-injection with mean long axis of 2.23mm and mean tumor volume of 4.17mm³. However, IVIS imaging was able to detect tumor growth at 14 days post-injection. Ascites formation was detected in mice at 21 days post-injection.

**Funding:** This work was supported by VeloSano Bike to Cure to OR. The funder had no role in study design, data collection and analysis, decision to publish, or preparation of the manuscript.

**Competing interests:** The authors have declared that no competing interests exist.

## Conclusions

TAUS is highly discriminatory for monitoring EOC in pre-clinical murine model, allowing for detection of tumor dimension as small as 2 mm and as early as seven days post-injection compared to IVIS. In addition, TAUS provides relevant information for ascites development and detection of multiple small metastatic tumor implants. TAUS provides an accurate and reliable method to detect and monitor IP EOC growth in mouse xenografts.

## Introduction

Epithelial ovarian cancer (EOC) is a leading cause of gynecologic cancer related mortality in women [1]. The five-year overall survival for women with EOC is poor since the majority of patients present with advanced and metastatic disease [2]. Additionally, although patients initially respond well to treatment with surgery and chemotherapy with carboplatin and paclitaxel, the vast majority of women will recur [4–8]. Ovarian carcinomas primarily undergo peritoneal dissemination, and are often associated with malignant ascites. This pattern of spread is associated with vague symptoms which leads to delays in diagnosis [3]. There is a significant unmet need for methods to facilitate early diagnosis of EOC and advance current therapeutic options.

Pre-clinical research utilizing EOC cell lines and patient-derived xenografts shows tremendous promise in advancing the current understanding of EOC carcinogenesis and therapeutics [9–15]. In longitudinal pre-clinical studies, the ability to detect tumor engraftment and sequentially assess tumor volume utilizing non-invasive techniques is essential to assessing tumor growth and treatment response. However, despite the existence of many cell lines that closely replicate human EOC at a cellular level, difficulty monitoring intraperitoneal tumor formation, growth and metastasis remains a major limitation in the execution of preclinical EOC studies [9–15].

In-vivo monitoring of EOC cell lines can be accomplished using several well- developed techniques including RFP, GFP, luciferase and ROSA reporter systems [16, 17]. Bioluminescence *in vivo* imaging system (IVIS) using luciferase reporter containing cell lines has been commonly utilized to track tumor growth over time, but this technique has limitations. This imaging technique involves injection of luciferin in conjunction with tumor cells, which is invasive and can initiate an inflammatory response [18]. Additionally, this technique only provides qualitative information regarding tumor progression [17]. The primary concern for use of IVIS is the necessity to use modified cell lines which have a tendency for genetic drift and phenotypic alterations. As such, use of IVIS for monitoring of patient-derived xenografts is not feasible [19,20]. In addition, a pertinent characteristic of EOC patients is the development of ascites throughout the progression of disease and the accuracy of luciferase is diminished in the presence of abdominal ascites as a result of dilution [21–24]. These studies support that IVIS imaging technology has limitations in ones ability to perform in-vivo experiments utilizing PDX and cell lines prone to ascites development, which are important in EOC research. Therefore, development of novel imaging strategies to study EOC in animal models is needed.

In clinical practice, transabdominal ultrasound (TAUS) is frequently utilized in the evaluation of women with gynecologic diseases, including EOC [25,26]. Despite being non-invasive, cost-effective and accurate, data for use of TAUS for monitoring of EOC in pre-clinical murine models is limited [27]. The objective of this study was to evaluate intraperitoneal tumor

engraftment and growth in the presence and absence of ascites in a pre-clinical murine model of EOC utilizing both TAUS and IVIS imaging.

## Methods

### Cell lines and lentiviral transformation of ID8 cells with luciferase vector

ID8 and ID8-VEGF syngeneic EOC cell lines were cultured in Dulbecco Modified Eagle Medium (DMEM) media containing heat inactivated 5% FBS (Atlas Biologicals Cat # F-0500-D, Lot F31E18D1) and 100 U/mL penicillin-streptomycin and 1% insulin/transferrin/selenium and grown under standard conditions. HEK 293T/17 (ATCC CRL-11268) cells were plated at 65% confluence in a 100 mm dish and cultured in 9 mL DMEM supplemented with heat inactivated 10% FBS (Atlas Biologicals Cat # F-0500-D, Lot F31E18D1)[15]. ID8 cells were subsequently transfected with luciferase containing construct pHIV-Luciferase #21375 4.5 μg (Addgene) to generate the ID8-luc cells. Briefly, 3 mL of the DMEM media was removed and ID8 cells were co-transfected with Lipofectamine 3000 (L3000015 Invitrogen) 35 μL of Plus reagent / 41 μL of Lipofectamine 3000, 3rd generation packaging vectors pRSV-REV #12253 4.3 μg, pMDG.2 #12259 4.3 μg, and pMDLg/pRRE #12251 4.3 μg (Addgene) and lentiviral vector directing expression of luciferase reporter pHIV-Luciferase #21375 4.5 μg (Addgene) in 3 mL of OptiMEM media. Following 8 hours of incubation, media of the 293T/T17 cultures was replaced and following 18 hours of incubation media containing viral particles were harvested and filtered through a 0.45 μm Durapore PVDF Membrane (Millipore SE1M003M00). Viral transfections were carried out over 72 hours ID8 parental cells and transduced cells were selected by their resistance to 2 μg/mL puromycin (MP Biomedicals 0219453910). Prior to use in this experiment, activity of luciferase promoter and tumor growth was confirmed in a pilot cohort of mice.

### Ethics statement

All studies were carried out in accordance with protocols approved by the Institutional and Animal Care and Use Committee (approval # 2018–2003) or the Institutional Biosafety Committee (approval # IBC0920) of the Cleveland Clinic Lerner Research Institute Biological Resource Unit.

### Mouse xenografts

Ten female C57Bl/6 mice were purchased from Jackson Laboratories (Bar Harbor, ME) at 6 weeks of age. After two weeks of acclimation, mice underwent IP injection of 300μL of either 5x10$^6$ ID8-luc (n = 5) or ID8-VEGF (n = 5) cells. Following cell injection mice were monitored daily for three days then every 4 days for the duration of the study. Additionally, mice were monitored weekly by TAUS and IVIS imaging for tumor progression. Mice reached endpoint criteria and immediate CO2 asphyxiation followed by cervical dislocation, when any of the following conditions were observed: tumor burden >1.5cm$^2$ by TAUS, tumor interfered with animals ability to eat or drink, 20% weight loss, palpation of tumor elicited a pain response, became unresponsive or death was imminent, exhibited respiratory difficulty or hypothermia, or any sign of outward distress such as hunched posture, ruffled fur, and reduced motility. All efforts were made to minimize suffering and distress including the use of dietary gel packs and placing the cage on a heated surface following anesthesia during recovery observation.

All mice were removed from the study at 60 days post cell injection. None of the mice died before meeting endpoint criteria or pre-determined 60-day endpoint of study.

## Tumor monitoring

Ultrasonography was performed using a Vevo2100 (VisualSonics) with an abdominal imaging package and MS550D probe (40Hz)(Fig 1). TAUS surveillance was initiated seven days following IP tumor injection. TAUS was performed every seven days until study endpoint at 67 days. Mice were anesthetized using isoflurane (DRE Veterinary) and placed in the supine position. Following the removal of abdominal hair using Nair (Church & Dwight Co. Inc.), sterile ultrasound gel was applied to the abdomen. TAUS was performed using Vevo2100 (VisualSonics) using the abdominal imaging package and MS550D probe (40Hz) (S1 Fig). Throughout the duration of TAUS imaging, murine heart rate was monitored and heated platform was utilized to minimize distress. For each timepoint, when each individual mouse underwent ultrasound, the abdomen was assessed for tumor in four locations based on abdominal quadrants: 1) right upper, 2) right lower, 3) left upper, 4) left lower. In order to provide specific information about tumor size discrimination with ultrasound, the individual size of each tumor implant was reported. Each animal served as its own control for assessing longitudinal tumor growth. Tumors were noted to be absent or present at each assessment, as well as tumor location by quadrant (data not shown). Tumor dimensions (length and width) were recorded and tumor volume was calculated using the formula: (Length*(Width$^2$))/2. Following mouse necropsy, tumors were submitted for H&E for confirmation. Study design is depicted in Fig 1.

## 2D IVIS imaging

Bioluminescence images were taken within 48 hours of ultrasound images with IVIS Lumina (PerkinElmer) using D-luciferin as previously described [24]. Mice received an IP injection of D-luciferin (Goldbio LUCK-1G, 150mg/kg in 150mL) under inhaled isoflurane anesthesia. Images were normalized (Living Image Software) with a minimum and maximum radiance of $7.5810^5$ and $5.3910^8$ photons/second/cm$^2$/steradian, respectively. All images were obtained with a 15 second exposure. Average luminescent intensity in photons per second/cm$^2$/steradian was calculated and reported for each mouse abdomen.

## 3D IVIS imaging

Upon endpoint, bioluminescence and x-ray images were taken using the IVIS Spectrum system (PerkinElmer). Mice were sedated with 2% isoflurane (DRE Veterinary) inhalation in an

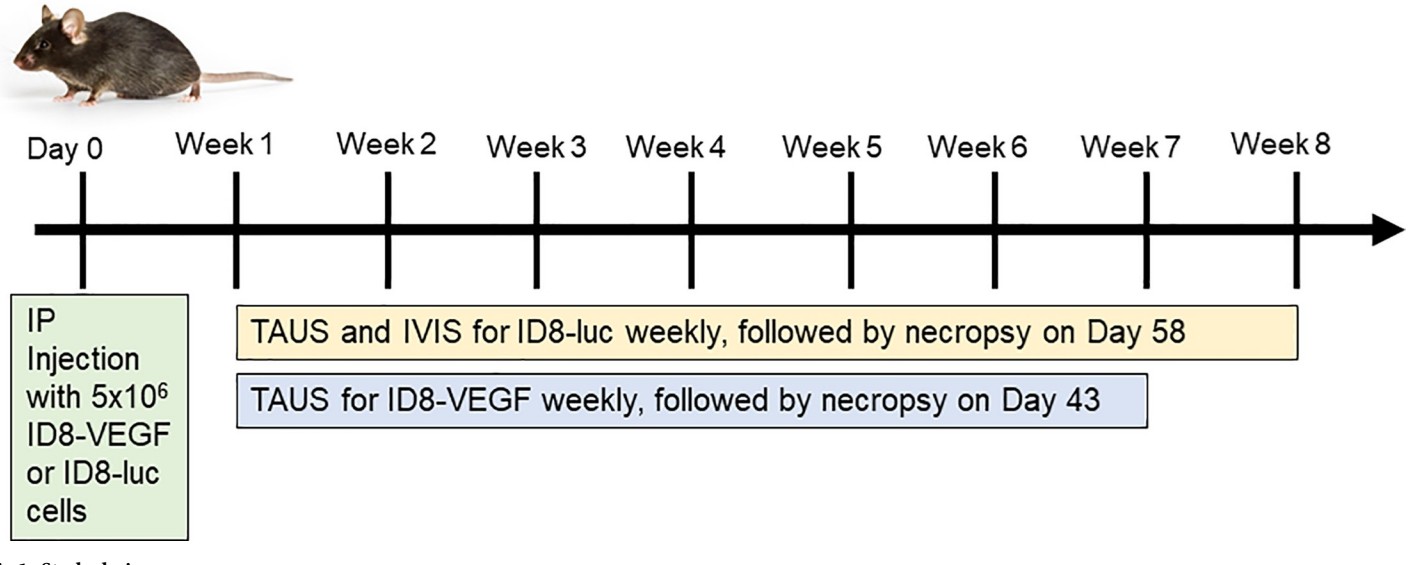

**Fig 1. Study design.**

airtight transparent anesthesia box for 5 minutes. Mice were shaved front and back and Nair was applied to remove remainder of the hair before being IP injected with D-luciferin (Gold-bio LUCK-1G, 150mg/kg in 150 mL). Mice are placed in a supine position on the light-tight chamber of the CCD camera imaging unit. Sequential images were acquired at 1min intervals (60 s exposure, no time delay) for at least 30 min. The luminescence camera was set to 60 s exposure, medium binning, f/1, blocked excitation filter, and open emission filter. The photographic camera was set to auto exposure, medium binning, and f/8. Average luminescent intensity in photons per second/cm$^2$/steradian was calculated and reported for each mouse abdomen. Identical settings were used to acquire each image and region of interest during the study. Ultrasound and IVIS imaging were performed independently by two separate investigators who were blinded to the results of the other imaging modality.

### Patient Derived Xenograft Pilot Study

In order to assess the utility of TAUS in PDX models of EOC, the investigators performed TAUS in the manner described above in a small cohort of mice (n = 5). In this pilot experiment, tumor samples were obtained at the time of primary debulking surgery for high grade serous EOC or uterine serous carcinoma, processed into a single-cell suspension and injected via IP injection. TAUS was done at four time points– 3 days, 10 days, 14 days and 21 days post-IP injection.

### Statistics

All data are presented as mean +/- standard deviation. Tumor volumes presented as mean+ SEM and graphed over time. All statistical analysis was performed in GraphPad Prism v8.

## Results

Tumor engraftment was detected in all C57Bl/6J via TAUS between 7–14 days, gross examination at necropsy and on histopathology. In addition, in mice injected with ID8-luc, tumor engraftment was noted at 14 days. In all cases, EOC tumors were detected before any clinical signs (ascites, palpable masses, lethargy). Beginning at Day 7 post-injection, TAUS was performed and a maximum of four tumor measurements were recorded per mouse in each abdominal quadrant. Six mice (60%) had one detectable tumor on TAUS at 7 days. All mice (n = 10) had at least one tumor detectable on TAUS at 14 days post-injection and 20% (n = 2) had two detectable tumors. Mean tumor dimensions and volumes for ID8-luc and ID8-VEGF are displayed in Table 1 and Table 2, respectively. The smallest tumor short and long axis measurements detected at 7 days were 1.74mm and 2.23mm, respectively. The lowest recorded tumor volume was 4.17mm. Ascites was detected as early as 21 days. Tumor volume detected via TAUS over time is displayed for both ID8 and ID8 VEGF mice in Fig 2. Fig 3 depicts weekly TAUS images of EOC tumor implant over time in ID8-luc without ascites (A) and in ID8-VEGF (B). To test whether TAUS can be used to detect PDX tumor, we injected mice (n = 5) with a PDX single cell suspension of EOC human cells. We were able to detect tumor growth for a human PDX tumor of high grade serous EOC at 10 days post injection (1.893mm in long axis). Longitudinal imaging demonstrated increased growth on days 14 (2.200mm in long axis) and 21 (3.875mm in long axis), respectively. (S2 Fig).

Within 48 hours of TAUS, 2D IVIS imaging was performed. ID8 tumor detection by 2D IVIS imaging was noted at 7 days post cell injection and intraperitoneal tumor growth over time was tracked as previously reported (Fig 4A and 4B). As the PDX and VEGF cell lines do not contain a luciferase reporter system, IVIS imaging was not performed on these cohorts. At Day 14 it was noted (Table 1) that the photon/second/cm^3/sr read was decreasing and then

**Table 1. Mean tumor measurements in ID8 mice with TAUS and IVIS imaging.**

| | Transabdominal Ultrasound Measurement | | | | IVIS Imaging | |
| | Mean Tumor Implant Long Axis (mm) (n = 20) | Mean Tumor Implant Short Axis (mm) (n = 20) | Mean Tumor Implant Volume (mm3) (n = 20) | Mean Total Tumor Burden per Mouse (n = 5) (mm3) | | photon/second/cm^2/sr |
|---|---|---|---|---|---|---|
| Day 7 | 2.83 | 1.80 | 4.61 | 4.61 | Day 7 | 2.63e6 |
| Day 14 | 2.92 | 2.11 | 6.81 | 9.55 | Day 14 | 1.11e6 |
| Day 21 | 4.10 | 2.58 | 14.5 | 43.38 | Day 24 | 4.15e6 |
| Day 28 | 4.26 | 2.76 | 17.28 | 69.10 | Day 31 | 4.88e6 |
| Day 35 | 4.94 | 3.50 | 32.54 | 130.14 | Day 39 | 7.75e6 |
| Day 42 | 5.88 | 4.14 | 51.34 | 205.35 | Day 46 | $1.95e^7$ |
| Day 49 | 7.20 | 4.58 | 75.58 | 305.53 | Day 57 | 2.18e7 |
| Day 56 | 8.56 | 5.06 | 110.00 | 432.83 | | |

increased at subsequent timepoints as expected. This speaks to the high variance of IVIS at early timepoints, and why having an added system of tumor progression analysis, such as TAUS, is essential. Additionally, 3D IVIS imaging including murine x-ray was performed at endpoint to determine tumor location (Fig 4C). All tumors were located in regions identified by 3D IVIS imaging at endpoint necropsy (data not shown).

Prior to necropsy, the murine abdominal cavity was imaged to confirm gross tumor presence in the ID8 and ID8 VEGF cohort. Each tumor was then excised and stained using H&E to confirm EOC histology (Fig 5). Following murine necropsy, ID8 tumors were imaged and excised (Fig 5). ID8 and ID8 VEGF EOC tumor phenotype was confirmed by histology (Fig 5A and 5B respectively).

## Discussion

Mouse xenografts represent an important method to pursue urgently needed preclinical studies to understand pathogenesis and develop new therapies for EOC. While many orthotopic

**Table 2. Tumor measurements in ID8-VEGF mice with transabdominal ultrasound.**

| | Transabdominal Ultrasound Measurement | | | | |
| | Mean Tumor Implant Long Axis (mm) (n = 20) | Mean Tumor Implant Short Axis (mm) (n = 20) | Mean Tumor Implant Volume (mm3) (n = 20) | Mean Total Tumor Burden per Mouse (n = 5) (mm3) | Presence of Ascites |
|---|---|---|---|---|---|
| Day 7 | 2.23 | 1.98 | 4.39 | 4.39 | No |
| Day 14 | 3.00 | 1.90 | 5.56 | 6.79 | No |
| Day 21 | 3.98 | 2.54 | 13.52 | 35.23 | Yes |
| Day 28 | 4.30 | 2.70 | 16.73 | 49.92 | Yes |
| Day 35 | 5.36 | 3.70 | 38.59 | 130.28 | Yes |
| Day 42 | 6.55 | 4.03 | 56.09 | 178.81 | Yes |

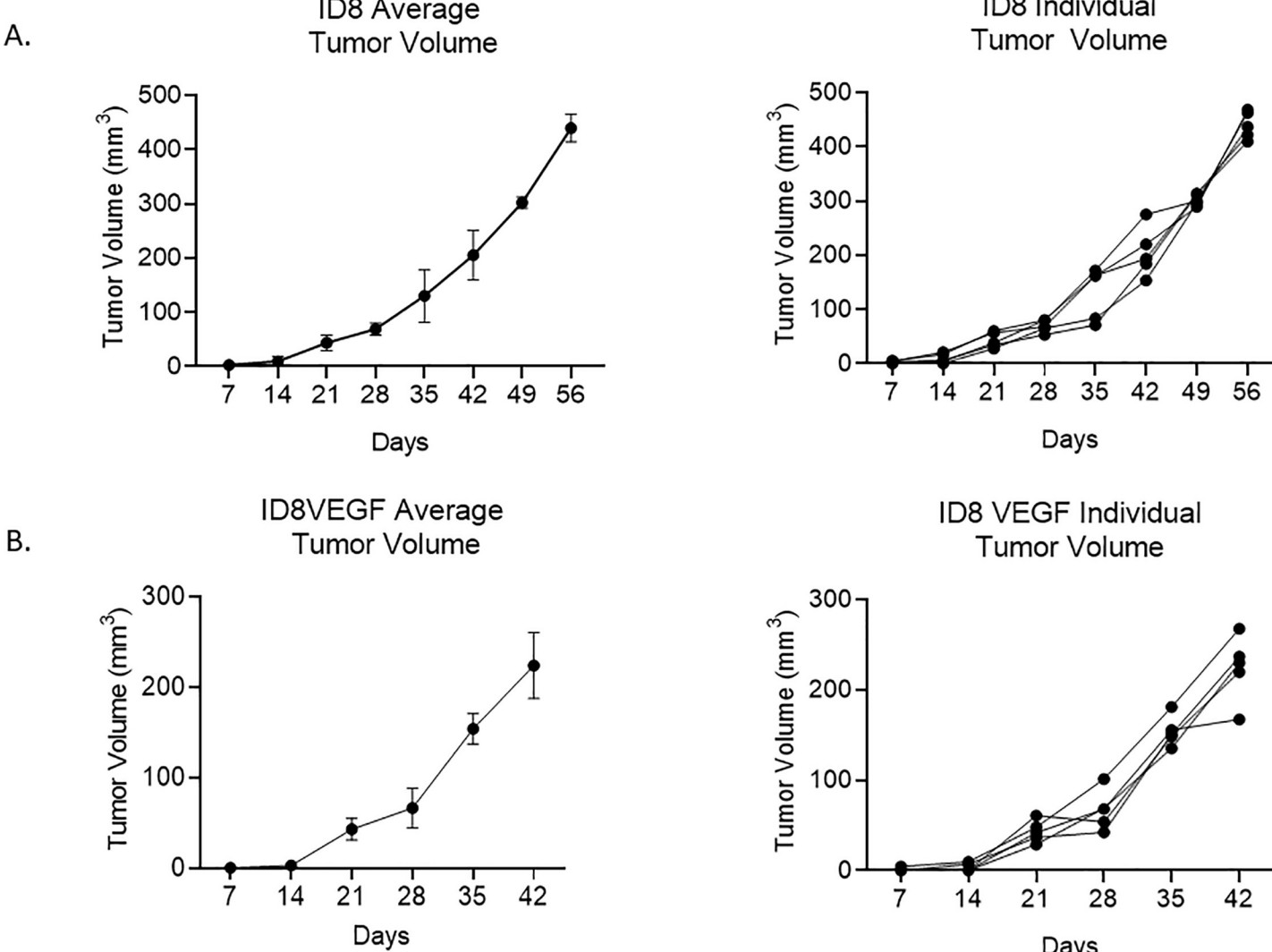

**Fig 2.** Transabdominal ultrasound allows for monitoring of tumor engraftment and growth in mice with ovarian cancer xenografts of ID8 (A) and ID8-VEGF (B).

models exist that closely mirror human EOC, techniques to monitor intraperitoneal tumors in an accurate, non-invasive fashion are limited. In this study, we applied ultrasonography, to evaluate the engraftment and growth of EOC in a pre-clinical model. We demonstrated that in murine models of EOC, TAUS can be used to accurately detect and monitor the growth of EOC xenografts with tumors and ascites detected as early as 7 and 21 days post-injection, respectively. We found TAUS is more sensitive for detection of disease progression compared to bioluminescence assays where tumor detection first occurred at 14 days post-injection. Our findings are consistent with prior studies demonstrating that IVIS is able to detect tumor growth at 2 weeks after ID8-luc cell injection [21].

Currently utilized and previously described strategies for tumor monitoring in murine models of EOC fall short [9–15, 21–24]. IVIS imaging is frequently used for tumor assessment in murine models of EOC but has significant limitations. First, the cell-line must contain a luciferase reporter, which limits the ability to utilize high fidelity patient-derived tumorgraft models. Second, concerns exist regarding initiation of an inflammatory response or other

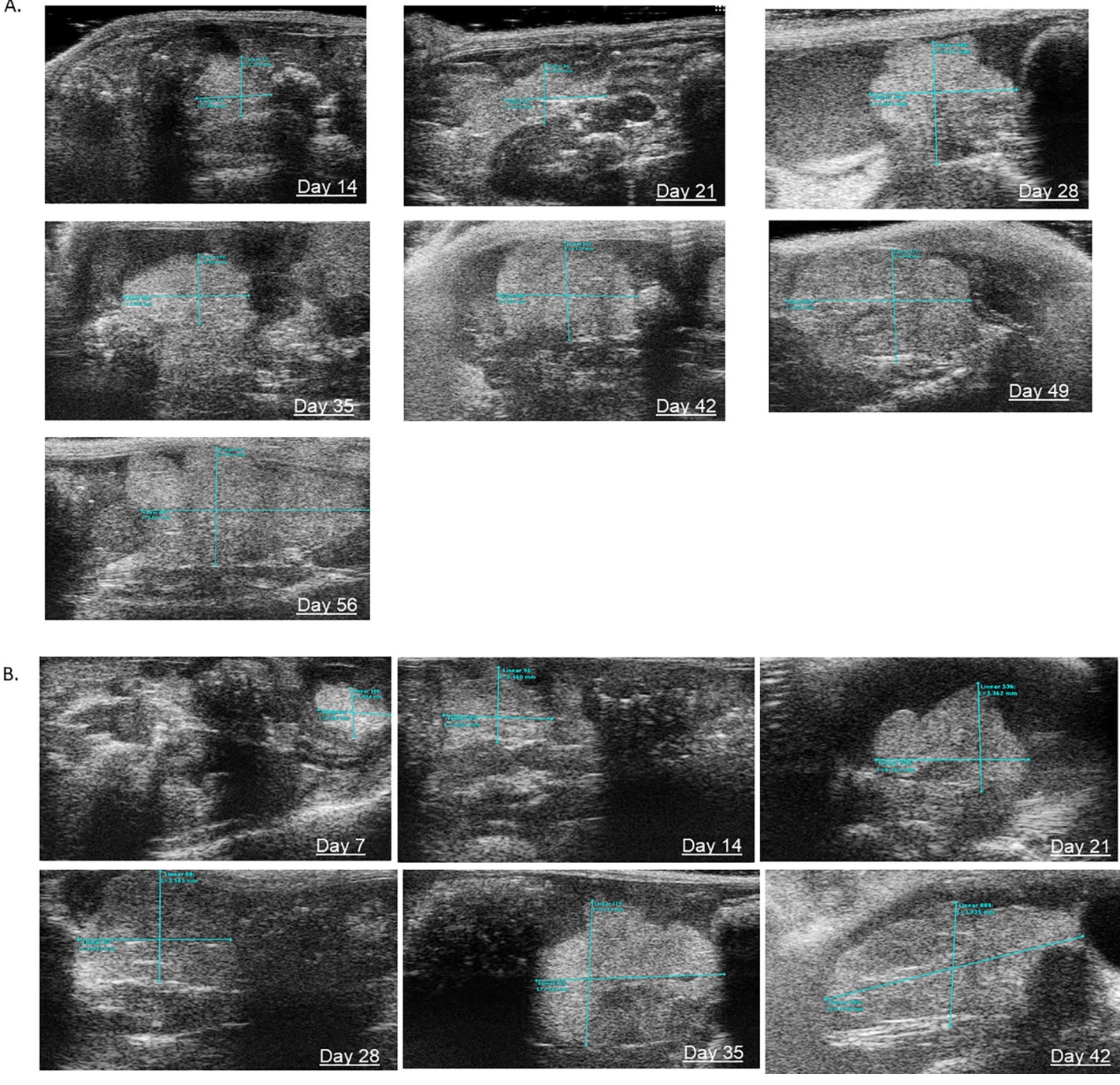

**Fig 3. Transabdominal ultrasound allows for longitudinal monitoring of intraperitoneal tumor implants in murine model of EOC.**

phenotypic and genotypic alterations that may render the cell line less applicable to human EOC [19–21,23]. Finally, detection of ascites is compromised in IVIS models. Baert et al demonstrated that reduced sensitivity of IVIS in the presence of luciferase with a significantly decreased in the presence of ascites within an ID8-luc model [23]. As the majority of human and mouse EOC lines have a penchant for ascites development, the detection of ascites is of high importance. In the clinical setting, ascites significantly impacts patient quality of life and

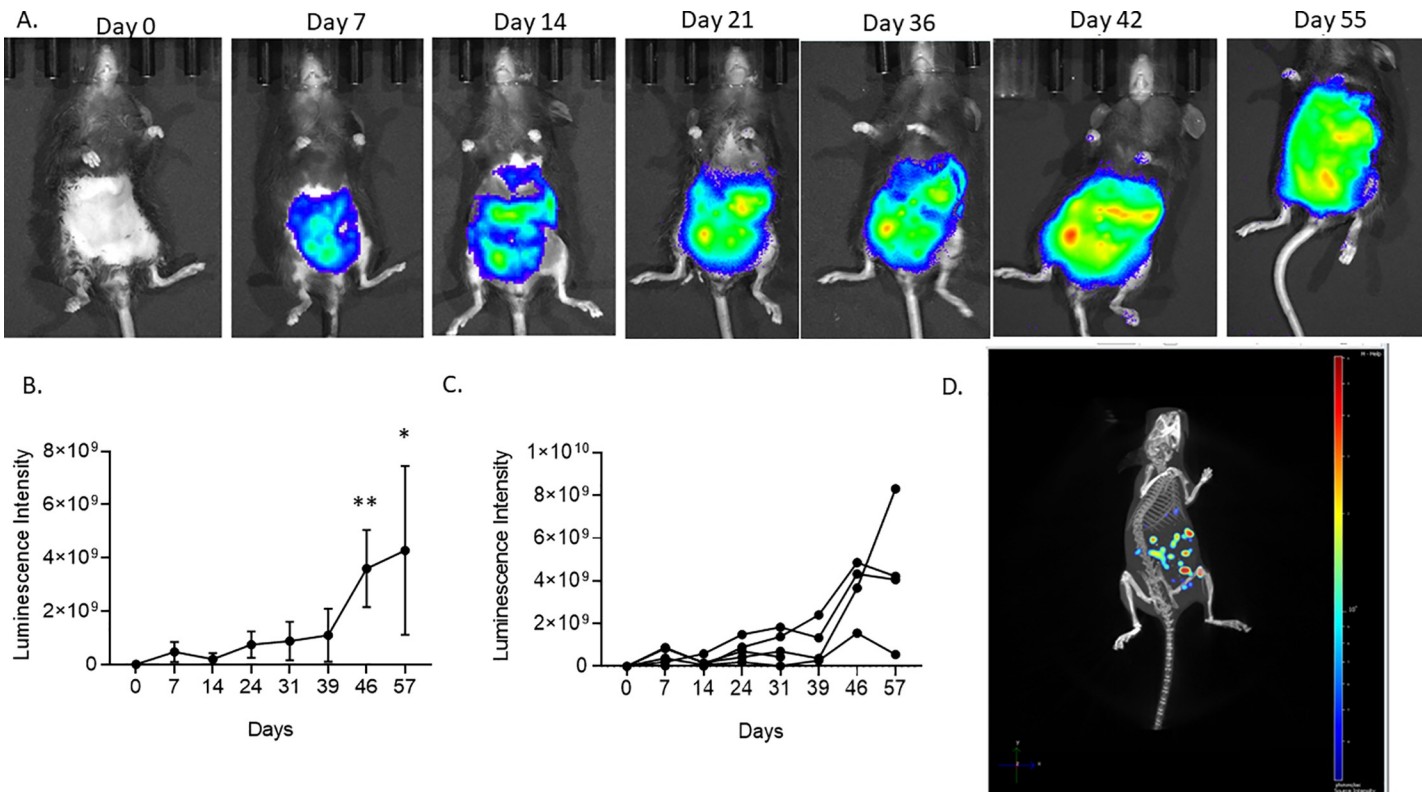

**Fig 4. 2D IVIS imaging tracked tumor growth over time, and 3D IVIS imaging determined endpoint tumor volume in ID8 tumor bearing C57Bl/6 mice.**

is a harbinger of advanced, progressive disease. Ascites is important to study in pre-clinical translational models as it can yield diagnostic and prognostic information.

In clinical practice, TAUS is frequently utilized in the evaluation of women with gyneco-logic diseases, including EOC [25,26]. However, prior to this study, application of ultrasonog-raphy to murine pre-clinical EOC models has been limited. Weroha et al. utilized ultrasonography to assess tumor growth in patient-derived xenografts of EOC with high corre-lation between ultrasound assessment and tumor measurements at necropsy [27]. In addition, TAUS has been utilized in pre-clinical models of non-gynecologic intra-abdominal cancers, including pancreatic and genitourinary malignancies [28–30]. Within a murine model of blad-der cancer, Patel et al. demonstrated high correlation between tumor size with transabdominal micro-ultrasound and at necropsy and were able to detect tumors as small as 0.95 mm$^3$ [30]. Similarly, in pre-clinical murine models of pancreatic adenocarcinoma, intra-pancreatic tumors were detected as early as three days post-injection, and tumor metastasis in addition to ascites was identified in all animals at two weeks with excellent correlation to necropsy tumor volume [29].

TAUS offers several potential advantages over currently available imaging tools for the monitoring of murine models of EOC. Primarily, we demonstrate in this study that tumor detection can be assessed as early as one week post-injection, with tumor implants detected as small as 2mm in longest dimension. Secondly, malignant ascites and innumerable tumor implants are pathognomonic of human EOC. This method allows researchers to monitor treatment response via tumor volume and ascites in parallel to patients undergoing chemo-therapy where radiologic scoring systems such as RECIST criteria are used. In addition, TAUS can be utilized for EOC monitoring in cell lines that do not have RFP, GFP, luciferase or

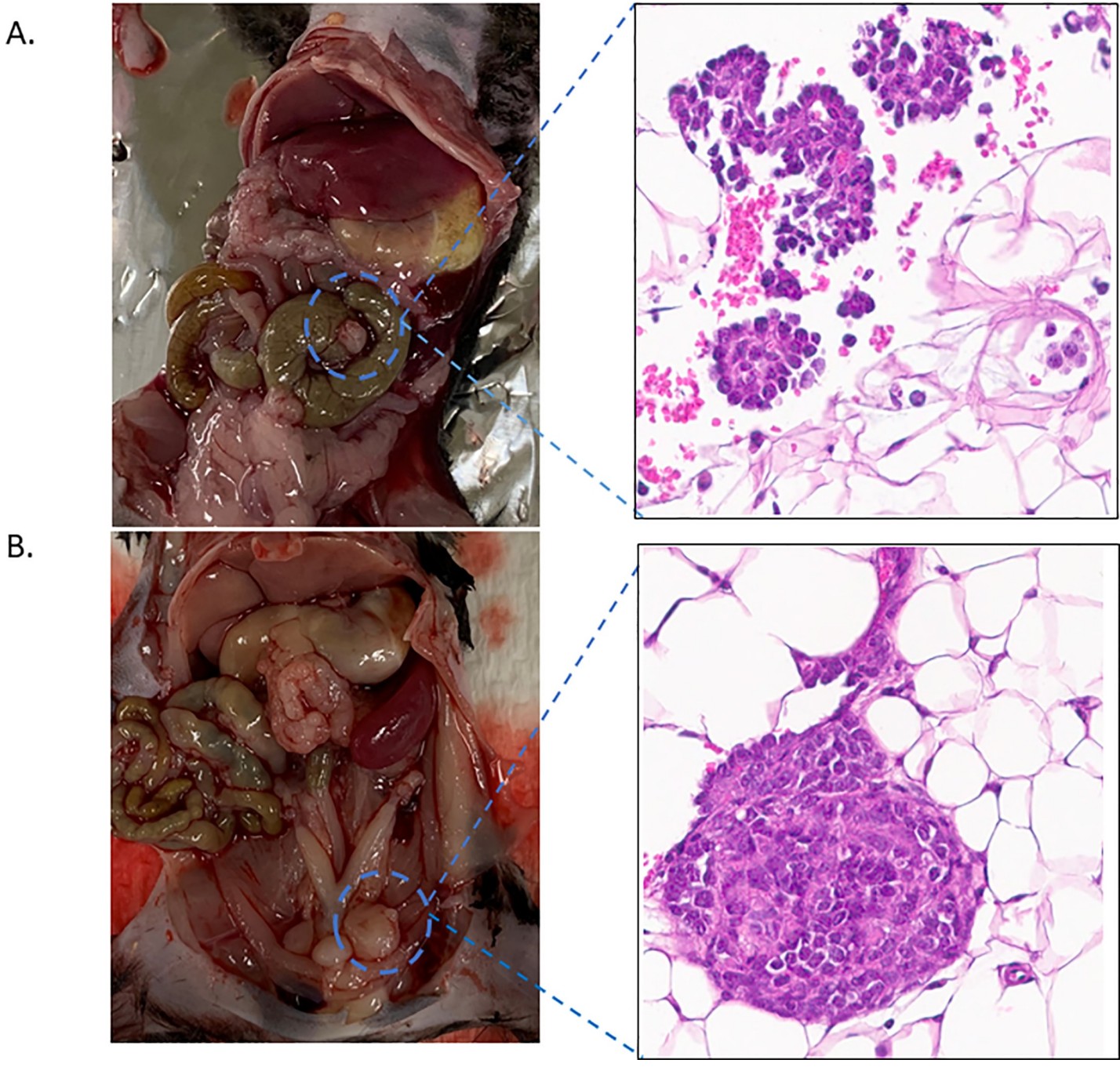

**Fig 5. Upon necropsy macroscopic and histologic EOC tumors were identified and validated for both ID8 and ID8 VEGF cell lines.**

ROSA reporter systems. Thereby, this allows for in-vivo monitoring of any intra-peritoneal EOC cell line with or without ascites development, including PDX models. This is important because it allows researchers to follow tumor growth and treatment response over time with cells transplanted directly from patient tumor specimens without the need for luciferase transduction. Finally, the ability to accurately detect tumors may represent a strategy to minimize animal euthanasia, as their disease burden can be monitored in-vivo to end-point during an

experiment without need for early necropsy with each animal serving as its own control. Therefore, monitoring EOC growth and response via TAUS has improved detection, higher sensitivity and increased breadth and utility over presently utilized imaging techniques.

In clinical practice, transabdominal and transvaginal US remain gold-standard for the initial assessment of gynecologic pathology, including ovarian tumors. TAUS is non-invasive and cost-effective with low risk to the patient. In this study, we demonstrate that this same imaging modality can be applied to mouse xenografts. Based on these results, we have adopted TAUS as a method to monitor tumor growth and treatment response in EOC preclinical studies in both syngeneic and PDX models with excellent success and reproducibility. One limitation of this model is the need for mouse anesthesia during TAUS. In this series, anesthesia and TAUS were well tolerated by the mice with no adverse intra-anesthesia events or mortalities related to the procedure. In our reported experience, the application and interpretation of TAUS imaging to murine models of EOC is feasible and is no more challenging than other imaging modalities utilized for pre-clinical tumor monitoring, including IVIS. Despite this, to the best of our knowledge, this study represents the first publication assessing the feasibility of TAUS for preclinical murine models of EOC in parallel with IVIS imaging.

In conclusion, TAUS shows promise in the detection of tumor growth and metastasis and response to therapies in intraperitoneal mouse xenografts of EOC. TAUS allows for detailed measurements of tumors and metastatic implants, ascites and is more sensitive than IVIS imaging.

## Supporting information

**S1 Fig. Procedural steps for transabdominal ultrasound.**
(TIF)

**S2 Fig. Human derived EOC PDX tumor detected at 10 days post IP injection (A).**
(TIF)

## Author Contributions

**Conceptualization:** Laura M. Chambers, Ofer Reizes.

**Data curation:** Laura M. Chambers, Emily Esakov, Chad Braley.

**Formal analysis:** Laura M. Chambers, Emily Esakov.

**Funding acquisition:** Laura M. Chambers, Chad Michener, Ofer Reizes.

**Investigation:** Emily Esakov, Ofer Reizes.

**Methodology:** Laura M. Chambers.

**Project administration:** Mariam AlHilli, Chad Michener, Ofer Reizes.

**Resources:** Ofer Reizes.

**Supervision:** Ofer Reizes.

**Writing – original draft:** Laura M. Chambers, Emily Esakov, Chad Braley.

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
