## [Decision Letter · Decision Letter 0]

5 Mar 2020

PONE-D-20-00914

Use of Transabdominal Ultrasound for the Detection of Intra-Peritoneal Tumor Engraftment and Growth in Mouse Xenografts of Epithelial Ovarian Cancer

PLOS ONE

Dear Dr. Reizes,

Thank you for submitting your manuscript to PLOS ONE. After careful consideration, we feel that it has merit but does not fully meet PLOS ONE’s publication criteria as it currently stands. Therefore, we invite you to submit a revised version of the manuscript that addresses the points raised during the review process.

We would appreciate receiving your revised manuscript by Apr 19 2020 11:59PM. To enhance the reproducibility of your results, we recommend that if applicable you deposit your laboratory protocols in protocols.io, where a protocol can be assigned its own identifier (DOI) such that it can be cited independently in the future. For instructions see: http://journals.plos.org/plosone/s/submission-guidelines#loc-laboratory-protocols

We look forward to receiving your revised manuscript.

Kind regards,

Shannon M. Hawkins, M.D., Ph.D.

Academic Editor

PLOS ONE

Journal Requirements:

2) In your Methods section, please give the sources of any cell lines used in your study (such as ID8 and ID8-VEGF).

3) Please include captions for your Supporting Information files at the end of your manuscript, and update any in-text citations to match accordingly. Please see our Supporting Information guidelines for more information: http://journals.plos.org/plosone/s/supporting-information.

4) Please include a title for Tables 1 and 2.

Reviewers' comments:

Reviewer's Responses to Questions

**Comments to the Author**

1. Is the manuscript technically sound, and do the data support the conclusions?

Reviewer #1: Partly

Reviewer #2: Yes

2. Has the statistical analysis been performed appropriately and rigorously? 

Reviewer #1: No

Reviewer #2: Yes

3. Have the authors made all data underlying the findings in their manuscript fully available?

Reviewer #1: Yes

Reviewer #2: Yes

4. Is the manuscript presented in an intelligible fashion and written in standard English?

Reviewer #1: No

Reviewer #2: Yes

5. Review Comments to the Author

Reviewer #1: The manuscript entitled “Use of Transabdominal Ultrasound for the Detection of Intra-Peritoneal Tumor Engraftment and Growth in Mouse Xenografts of Epithelial Ovarian Cancer” by Chambers et al. have described the results detecting tumor development in ID8 mouse EOC cell injected mice using both transabdominal ultrasound (TAUS) and bioluminescence in vivo imaging system

(IVIS). The time-dependent detection limits reported are informative. However, this study suffers several major flaws.

1. In Introduction, the authors stated “The objective of this study was to evaluate intraperitoneal tumor engraftment and growth in the presence and absence of ascites in a pre-clinical murine model of EOC utilizing both TAUS and IVIS imaging.” This goal has a very limited scale and significance to ovarian cancer research. In particular, the IVIS has been used for mouse imaging for many years.

2. Two areas need to be included in this study to have enough significant impact and be suitable to the scale of a PlosOne publication: 1) include a human xenograft model(s) to test the detection limitation of human tumors in mice; and 2) include several known EOC markers in blood/ascites/or peritoneal washing to compare to TAUS sensitivity and specificity (by including a few control mice) in detecting early stage of EOC.

3. In particular, Much16, HE4, and VEGF in blood/ascites/or peritoneal washing should be measured at the same time points with TAUS and IVUS to compare their sensitivity and specificities.

4. Control mice are needed for TAUX and IVIS detection.

5. Importantly, the 10 mice used in study only sacrificed by the end to confirm the tumors. The TAUX imagines shown in Fig. 2 indicate that it is difficult to quantify tumors using TAUX. Validation of early-detected tumors and their measurements are critical for the study.

6. The detection ability related to tumor locations should be described.

7. The ID8-VEGF cells are much more aggressive as reported (PMID: 16886597). However, in this study, no difference or even less tumor burdens in the ID8-VEGF cell injected mice from Day 14 to Day 42. This needs to be clearly interpreted.

8. Why did the photon/second/cm^3/sr at Day 14 was reduced from Day 7 (Table 1) needs to be explained.

9. Tables 1-2 should have titles.

Reviewer #2: 1. IVIS 2D imaging, 15 sec exposure seems long even for the earlier time points. Did the author try different time of exposure?

2. Table 1-under IVIS imaging, the right part of the table, the intensity should be shown as cm^2 if it’s 2D imaging.

3. Tables 1 and 2- what does the n=20 mean under “Transabdominal ultrasound measurement”

4. Figure 3, image from day 7 is missing and should be included.

5. Consider including a figure illustrating the study design

6. Provide more details about the PDX and the PDX study- patient information, tumor information, number of cells injected, number of mice injected and any other time points assessed using TAUS or was only day 10. Did ref 27 also examine pre-necropsy time points?

7. Are there other IVIS imaging studies using the ID8-luc model that the authors can directly compare their results at similar time points?

8. Line 66: IVIS not IVUS

6. PLOS authors have the option to publish the peer review history of their article (what does this mean?). If published, this will include your full peer review and any attached files.

Reviewer #1: No

Reviewer #2: No

---

## [Author Response · Author response to Decision Letter 0]

5 Apr 2020

Manuscript: PONE-D-20-00914

Title: Use of Transabdominal Ultrasound for the Detection of Intra-Peritoneal Tumor Engraftment and Growth in Mouse Xenografts of Epithelial Ovarian Cancer

Corresponding Author: Ofer Reizes PhD; First Author: Laura Chambers, DO

Response to Reviewers

Reviewer #1: 

The manuscript entitled “Use of Transabdominal Ultrasound for the Detection of Intra-Peritoneal Tumor Engraftment and Growth in Mouse Xenografts of Epithelial Ovarian Cancer” by Chambers et al. have described the results detecting tumor development in ID8 mouse EOC cell injected mice using both transabdominal ultrasound (TAUS) and bioluminescence in vivo imaging system

(IVIS). The time-dependent detection limits reported are informative. However, this study suffers several major flaws.

1. In Introduction, the authors stated “The objective of this study was to evaluate intraperitoneal tumor engraftment and growth in the presence and absence of ascites in a pre-clinical murine model of EOC utilizing both TAUS and IVIS imaging.” This goal has a very limited scale and significance to ovarian cancer research. In particular, the IVIS has been used for mouse imaging for many years.

Response: While IVIS has been utilized for imaging in mouse models of EOC, this method does have limitations. As discussed in the introduction, IVIS necessitates a modified cell line with a promoter and studies have demonstrated that this may induce an inflammatory response. Specifically, for EOC research, IVIS is unable to be used in animal studies of patient-derived xenografts (PDX), which provide important pre-clinical data that cell lines cannot [19,20]. In addition, EOC tumors are prone to ascites development. Ascites in EOC significantly impacts survival outcomes and patient quality of life. In the presence of ascites, the accuracy of IVIS is significantly diminished due to dilution of substrate [21-24]. In our practice, we have encountered that IVIS imaging technology falls short in our ability to perform in-vivo murine experiments utilizing PDX and cell lines without promoters, especially in the early detection of tumors prior to chemotherapy initiation. The authors believe there is an un-met need to develop further imaging strategies to study EOC in animal models. Ultrasound is used widely in clinical practice for the diagnosis and management of gynecologic pathology, including ovarian cancer, and in translation research in other malignancies, including genitourinary malignancies. Therefore, the authors set out to investigate whether ultrasound can be similarly used in murine models of EOC. This has been further explained in the introduction. 

Line Number(s): 125-128

Revised Text: “These studies support that IVIS imaging technology has limitations in the ability to perform in-vivo experiments utilizing PDX and cell lines prone to ascites development, which are important in EOC research. Therefore, development of alternative imaging strategies to study EOC in animal models is needed.”

2. Two areas need to be included in this study to have enough significant impact and be suitable to the scale of a PlosOne publication: 1) include a human xenograft model(s) to test the detection limitation of human tumors in mice; and 2) include several known EOC markers in blood/ascites/or peritoneal washing to compare to TAUS sensitivity and specificity (by including a few control mice) in detecting early stage of EOC.

Response: The authors performed a pilot study utilizing our gynecologic cancer biorepository with human PDX models of EOC and uterine serous carcinoma (n=5). In this experiment, mice underwent TAUS at four time points (3 days, 10 days, 14 days, 21 days), to assess the utility of TAUS in detection. Tumors were able to be detected as early as 10 days utilitizing a high grade serous EOC line in a patient who underwent primary debulking surgery for Stage IIIC disease. This has been described within the methods and results below. 

Line Number(s): 207-213; 235-238

Revised Text: “In order to assess the utility of TAUS in PDX models of EOC, the investigators performed TAUS in the manner described above in a small cohort of mice (n=5). In this pilot experiment, tumor samples were obtained at the time of primary debulking surgery for high grade serous EOC or uterine serous carcinoma, processed into a single-cell suspension and injected via IP injection. TAUS was done at four time points – 3 days, 10 days, 14 days and 21 days post-IP injection.

“We were able to detect tumor growth for a human PDX tumor of high grade serous EOC at 10 days post injection (1.893mm in long axis). Longitudinal imaging demonstrated increased growth on days 14 (2.200mm in long axis) and 21 (3.875mm in long axis), respectively (Supplemental Figure 1).”

3. In particular, Much16, HE4, and VEGF in blood/ascites/or peritoneal washing should be measured at the same time points with TAUS and IVUS to compare their sensitivity and specificities.

Response: Assessment of MUC16, HE4 and VEGF in ascites and blood have demonstrated tremendous promise in the assessment of treatment response in both patients and animal models of EOC. Unfortunately, the role of these biomarkers is limited in the assessment of early stage EOC, especially prior to ascites and metastasis formation. In our study, ascites development was not detected until 21 days post tumor injection, compared to TAUS which detected ultrasound at 7 days. Specifically, high levels of HE4 in ascites has been linked to chemo-resistance (Liu) and may predict for optimal debulking in primary surgery (Paunovic) and overall survival (Braicu), but there is no data in the literature supporting HE4 should be used as a diagnostic biomarker of early stage ovarian cancer. The authors believe that the lack of useful biomarkers and imaging to detect early stage EOC supports how TAUS, which detects tumors as early as 7 days post-injection, may be a helpful tool in pre-clinical models. In addition, TAUS detects disease prior to the development of ascites and may be useful for study of metastasis, and response to chemotherapy in early stage disease. 

Line Number(s): N/A

Revised Text: N/A

References: 

1. Braicu EI, Fotopoulou C, Van Gorp T, Richter R, Chekerov R, Hall C, et al. Preoperative HE4 expression in plasma predicts surgical outcome in primary ovarian cancer patients: results from the OVCAD study. Gynecol Oncol. 2013 Feb;128(2):245-51.

2. Liu D, Kong D, Li J, Gao L, Wu D, Liu Y, et al. HE4 level in ascites may assess the ovarian cancer chemotherapeutic effect. J Ovarian Res. 2018 Jun 14;11(1):47,018-0402-3.

3. Paunovic V, Protrka Z, Ardalic D, Paunovic T. Usefulness of human epididymis protein 4 in predicting optimal cytoreductive therapy in patients with advanced ovarian cancer. J BUON. 2017 Jan-Feb;22(1):29-33

4. Control mice are needed for TAUX and IVIS detection.

Response: The authors did not use control animals in this study. Primarily, in this study design, each mouse serves as its own control throughout the study, which is parallel to clinical practice where individual tumors are followed and assessed for response. Secondly, based on previous studies validating use of transabdominal ultrasound for genitourinary and gastrointestinal tumors, control animals without tumors were not utilized as each animal was used as its own control [28-30]. In addition, the authors aimed to use the lowest number of animals to complete the study objectives per our institutional IACUC governance. 

Line Number(s): 178-179

Revised Text: “Each animal served as its own control for assessing longitudinal tumor growth.”

5. Importantly, the 10 mice used in study only sacrificed by the end to confirm the tumors. The TAUX imagines shown in Fig. 2 indicate that it is difficult to quantify tumors using TAUX. Validation of early-detected tumors and their measurements are critical for the study.

Response: Previous studies have demonstrated that transabdominal ultrasound measurements are comparable to weights of tumors at necropsy. The objective of this study was to specifically correlate tumor measurements from IVIS and TAUS for preclinical models of EOC. In this study, we identified tumor growth as early as seven days post-injection, which is comparable to studies in gastrointestinal and genitourinary malignancies. In our results, IVIS was able to detect tumors at day 14 post-injection, which is similar to previously published data in ID8-luc cell lines. In our experience and with appropriate training of laboratory staff, we have found that the application and interpretation of TAUS imaging to murine models of EOC is feasible and is no more challenging than other imaging modalities utilized for pre-clinical tumor monitoring, including IVIS. 

Line Number(s): 334-337

Revised Text: “. In our reported experience, the application and interpretation of TAUS imaging to murine models of EOC is feasible and is no more challenging than other imaging modalities utilized for pre-clinical tumor monitoring, including IVIS.”

6. The detection ability related to tumor locations should be described.

Response: There is no difference in ability to detect tumors based on location with TAUS. 

Line Number(s): N/A

Revised Text: N/A

7. The ID8-VEGF cells are much more aggressive as reported (PMID: 16886597). However, in this study, no difference or even less tumor burdens in the ID8-VEGF cell injected mice from Day 14 to Day 42. This needs to be clearly interpreted.

Response: Although the ID8 VEGF does not exhibit enhanced rate of tumor growth, the debilitating ascites that develops in these mice speaks to the aggressive nature of the subtype of EOC. As such, ID8 VEGF mice had overall poorer health when compared to the ID8 cohort and were removed from the study at an earlier timepoint. 

Line Number(s): Table 2

Revised Text: N/A

8. Why did the photon/second/cm3/sr at Day 14 was reduced from Day 7 (Table 1) needs to be explained.

Response: WE thank the reviewers for pointing this out and have addressed it in the text.

Line Number(s): 249-252

Revised Text: At Day 14 it was noted (Table 1) that the photon/second/cm3/sr read was decreasing and then increased at subsequent timepoints as expected. This speaks to the high variance of IVIS at early timepoints, and why having an added system of tumor progression analysis, such as TAUS, is essential.

9. Tables 1-2 should have titles.

Response: Titles have been added. 

Line Number(s): Table 1/2

Revised Text: N/A

 

Reviewer #2: 

1. IVIS 2D imaging, 15 sec exposure seems long even for the earlier time points. Did the author try different time of exposure?

Response: The 2D IVIS system has an automatic exposure setting of 0.5-60seconds determined by the initially snapshots collected to maintain the signal within the linear range and prevent oversaturation. To account for this, before analysis all images were manually adjusted to the same minimum and maximum luminescent scales, as standard manufacturer operating procedure details.

Line Number(s): N/A

Revised Text:N/A

2. Table 1-under IVIS imaging, the right part of the table, the intensity should be shown as cm2 if it’s 2D imaging.

Response: This has been amended. 

Line Number(s): Table 1

Revised Text: N/A

3. Tables 1 and 2- what does the n=20 mean under “Transabdominal ultrasound measurement”

Response: For each timepoint, when each individual mouse underwent ultrasound, the abdomen was assessed for tumor in four locations based on abdominal quadrants: 1) right upper, right lower, left upper, left lower. In order to provide specific information about tumor size discrimination with ultrasound, we reported the individual size of each tumor implant. This has been further clarified within the methods. 

Line Number(s): 174-178

Revised Text: “For each timepoint, when each individual mouse underwent ultrasound, the abdomen was assessed for tumor in four locations based on abdominal quadrants: 1) right upper, 2) right lower, 3) left upper, 4) left lower. In order to provide specific information about tumor size discrimination with ultrasound, the individual size of each tumor implant was reported.”

4. Figure 3, image from day 7 is missing and should be included.

Response: This has been added to the figure.

Line Number(s): N/A

Revised Text:Figure 3

5. Consider including a figure illustrating the study design

Response: A figure explaining study design has been added. 

Line Number(s): Figure 1

Revised Text: Figure 1 

6. Provide more details about the PDX and the PDX study- patient information, tumor information, number of cells injected, number of mice injected and any other time points assessed using TAUS or was only day 10. Did ref 27 also examine pre-necropsy time points?

Response: Imaging of mice with PDX tumors were performed as a pilot study to demonstrate whether tumors could be detected. Details of this have been expanded in the methods section. 

Line Number(s): 207-213. 

Revised Text: “In order to assess the utility of TAUS in PDX models of EOC, the investigators performed TAUS in the manner described above in a small cohort of mice (n=5). In this pilot experiment, tumor samples were obtained at the time of primary debulking surgery for high grade serous EOC or uterine serous carcinoma, processed into a single-cell suspension and injected via IP injection. TAUS was done at four time points – 3 days, 10 days, 14 days and 21 days post-IP injection.”

7. Are there other IVIS imaging studies using the ID8-luc model that the authors can directly compare their results at similar time points?

Response: Our bioluminescence results for IVIS are comparable to those reported in Liao et al [21] utilizing ID8-luciferase cells with both studies detecting tumor growth at 2 weeks post-injection. In their study, they reported that bioluminescence imaging was able to detect tumor growth and engraftment earlier than performing mouse weight measurements, alone. 

Line Number(s): 262-264

Revised Text: “Our findings are consistent with prior studies demonstrating that IVIS is able to detect tumor growth at 2 weeks after ID8-luc cell injection [21].”

8. Line 66: IVIS not IVUS

Response: This changed has been made. 

Line Number(s): Line 66

Revised Text: “TAUS is highly discriminatory for monitoring EOC in pre-clinical murine model, allowing for detection of tumor dimension as small as 2 mm and as early as seven days post-injection compared to IVIS.”

---

## [Decision Letter · Decision Letter 1]

10 Apr 2020

Use of Transabdominal Ultrasound for the Detection of Intra-Peritoneal Tumor Engraftment and Growth in Mouse Xenografts of Epithelial Ovarian Cancer

PONE-D-20-00914R1

Dear Dr. Reizes,

We are pleased to inform you that your manuscript has been judged scientifically suitable for publication and will be formally accepted for publication once it complies with all outstanding technical requirements.

With kind regards,

Shannon M. Hawkins, M.D., Ph.D.

Academic Editor

PLOS ONE

Additional Editor Comments (optional):

Reviewers' comments:

Reviewer's Responses to Questions

**Comments to the Author**

1. If the authors have adequately addressed your comments raised in a previous round of review and you feel that this manuscript is now acceptable for publication, you may indicate that here to bypass the “Comments to the Author” section, enter your conflict of interest statement in the “Confidential to Editor” section, and submit your "Accept" recommendation.

Reviewer #2: All comments have been addressed

2. Is the manuscript technically sound, and do the data support the conclusions?

Reviewer #2: Yes

3. Has the statistical analysis been performed appropriately and rigorously? 

Reviewer #2: Yes

4. Have the authors made all data underlying the findings in their manuscript fully available?

Reviewer #2: Yes

5. Is the manuscript presented in an intelligible fashion and written in standard English?

Reviewer #2: Yes

6. Review Comments to the Author

Reviewer #2: The authors addressed previous concerns by including additional data and requested information.

I am satisfied with the revised version.

7. PLOS authors have the option to publish the peer review history of their article (what does this mean?). If published, this will include your full peer review and any attached files.

Reviewer #2: No

---

## [Editor Report · Acceptance letter]

15 Apr 2020

PONE-D-20-00914R1 

Use of Transabdominal Ultrasound for the Detection of Intra-Peritoneal Tumor Engraftment and Growth in Mouse Xenografts of Epithelial Ovarian Cancer 

Dear Dr. Reizes:

I am pleased to inform you that your manuscript has been deemed suitable for publication in PLOS ONE. Congratulations! Your manuscript is now with our production department. 

With kind regards,

on behalf of

Dr. Shannon M. Hawkins 

Academic Editor

PLOS ONE